Deep conservation of prion-like composition in the eukaryotic prion-former Pub1/Tia1 family and its relatives

Su Wan-Chun
Harrison Paul M. paul.harrison@mcgill.ca
Department of Biology, McGill University , Montreal, QC , Canada
Gillespie Joseph
Electronic publication date: 2020 Apr 17
Publication date: 2020
Volume: 8
Electronic Location ID: e9023
Received 2019 Nov 5; Accepted 2020 Mar 30
Copyright: © 2020 Su and Harrison
Copyright year: 2020
Copyright holder: Su and Harrison
License: This is an open access article distributed under the terms of the Creative Commons Attribution License, which permits unrestricted use, distribution, reproduction and adaptation in any medium and for any purpose provided that it is properly attributed. For attribution, the original author(s), title, publication source (PeerJ) and either DOI or URL of the article must be cited.
License URL: https://creativecommons.org/licenses/by/4.0/

Keywords: Prion, Pub1, Tia1, Nam8, RNA-binding, Compositional bias, Intrinsic disorder, Yeast, Human, Evolution

Funding: Natural Sciences Engineering Research Council of Canada Canada Foundation for Innovation Compute Canada This research was funded completely by the Natural Sciences Engineering Research Council of Canada, except that some of the calculations were performed on a desktop computer funded by both the Canada Foundation for Innovation and the Natural Sciences Engineering Research Council of Canada. Computing resources supplied by Compute Canada were also used. The funders had no role in study design, data collection and analysis, decision to publish, or preparation of the manuscript.

==============================
Pub1 protein is an important RNA-binding protein functional in stress granule assembly in budding yeast Saccharomyces cerevisiae and, as its co-ortholog Tia1, in humans. It is unique among proteins in evidencing prion-like aggregation in both its yeast and human forms. Previously, we noted that Pub1/Tia1 was the only protein linked to human disease that has prion-like character and and has demonstrated such aggregation in both species. Thus, we were motivated to probe further into the evolution of the Pub1/Tia1 family (and its close relative Nam8 and its orthologs) to gain a picture of how such a protein has evolved over deep evolutionary time since the last common ancestor of eukaryotes. Here, we discover that the prion-like composition of this protein family is deeply conserved across eukaryotes, as is the prion-like composition of its close relative Nam8/Ngr1. A sizeable minority of protein orthologs have multiple prion-like domains within their sequences (6–20% depending on criteria). The number of RNA-binding RRM domains is conserved at three copies over >86% of the Pub1 family (>71% of the Nam8 family), but proteins with just one or two RRM domains occur frequently in some clades, indicating that these are not due to annotation errors. Overall, our results indicate that a basic scaffold comprising three RNA-binding domains and at least one prion-like region has been largely conserved since the last common ancestor of eukaryotes, providing further evidence that prion-like aggregation may be a very ancient and conserved phenomenon for certain specific proteins.

Introduction

Prions in eukaryotes have been linked to diseases, evolutionary capacitance, large-scale genetic control and long-term memory formation. Prion formation and propagation have been studied extensively in the budding yeast Saccharomyces cerevisiae. The yeast S. cerevisiae has >200 prion-like proteins that have N/Q-rich domains of the sort observed in >10 known prion-formers (An, Fitzpatrick & Harrison, 2016). A domain of the yeast protein Pub1 exhibits prion-like protein aggregation and propagation in S. cerevisiae (Alberti et al., 2009). Pub1 in S. cerevisiae is a poly-(A)+-binding protein that functions in stress granule assembly, and is required for mRNA stability for many transcripts (Anderson, Paddy & Swanson, 1993; Matunis, Matunis & Dreyfuss, 1993; Rayman & Kandel, 2017). Pub1 protein abundance is linked to increased DNA replication stress (Tkach et al., 2012). There is evidence that the N/Q-rich prion-like domain of Pub1 is functional in recruiting protein synthesis machinery to the cytoskeleton, which is critical for maintaining cytoskeletal integrity (Li et al., 2014). In doing so, Pub1 interacts with the (PSI+)-prion-forming Sup35 protein to form a two-protein self-propagating state (Li et al., 2014). Pub1 can act as an accessory translation factor that appears to fine-tune translation termination efficiency (Urakov et al., 2017). Similarly to Sup35 (Franzmann et al., 2018), Pub1 forms condensates upon cellular stress (Kroschwald et al., 2018). Condensates induced by different stresses have different biophysical characteristics, i.e., some are gel-like, while others are more solid-like (Kroschwald et al., 2018). Another dimension to the relationship between Sup35 and Pub1 is that the latter appears to be involved in a cellular mechanism to alleviate toxicity of the (PSI+) prion, which is formed from Sup35 proteins (Urakov et al., 2018). Pub1 is one of a small number of prion-formers that serve as protein interaction hubs for other prion-like proteins (Harbi & Harrison, 2014b). The Pub1 prion-forming domain is one of the most rapidly evolving prion-forming domains sequence-wise (Su & Harrison, 2019). The Pub1 paralog Nam8 (which also contains a prion-like domain) is a part of the U1 snRNP protein complex; mutants of Nam8 are defective in meiotic recombination (Puig et al., 1999; Qiu, Schwer & Shuman, 2011). Nam8 was one of a small panel of genes that were discovered in a genome-wide genetic screen of yeast to rescue the cytotoxicity of FUS/TLS, a prion-like protein that causes some familial amyotrophic lateral scleroses (Ju et al., 2011). Nam8 is non-essential in the U1 snRNP protein complex, but demonstrates synthetic-lethal genetic interactions with other non-essential interactors (Agarwal, Schwer & Shuman, 2019).

The Pub1 human homologue, Tia1, is also functional through prion-like aggregation in stress granule assembly in mammalian cells (Gilks et al., 2004; Rayman & Kandel, 2017). The prion-like domain of Tia1 can be substituted with that of Sup35 and still maintain its stress granule assembly role, and forms protease-resistant aggregates in mammalian cells (Gilks et al., 2004). Full-length Tia1 can form self-perpetuating prion-like aggregates heterologously in S. cerevisiae (Li et al., 2014). Tia1 and its paralog Tiar (also called TiaL1) together regulate stress response and cell-cycle progression mediated by the protein kinase EIF2AK2 (Meyer et al., 2018). Reduction of Tia1 levels protects against tau-mediated neurodegeneration in the brain, rescues behavioural deficits and improves survival (Apicco et al., 2018). Single-nucleotide polymorphisms (SNPs) in Tia1 are linked at very low frequency to amyotrophic lateral sclerosis and frontotemporal dementia in Chinese cohorts but not in European ones (Baradaran-Heravi et al., 2018; Gu et al., 2018; Zhang et al., 2018). SNPs in Tia1 are linked to a multisystem proteinopathy that involves disruption of stress granule formation (Lee et al., 2018), and to Welander distal myopathy (Hackman et al., 2013; Klar et al., 2013).

All of these proteins (Pub1, Nam8, etc.) are examples of proteins that contain RNA-binding RRM domains. The RRM domain provides RNA-binding functionality for a diverse set of proteins across the eukaryotic clade, including dozens in S. cerevisiae. It comprises a very conserved alpha–beta protein fold that can bind to a large number of diverse RNA sequences (Daubner, Clery & Allain, 2013).

Previously, we noted that Pub1/Tia1 was the only protein linked to human disease that has prion-like character and aggregation in both humans and S. cerevisiae (An & Harrison, 2016). Here, motivated by these observations, we have probed further into the evolution of the Pub1/Tia1 family, including the close relative Nam8 and its orthologs. We discover that prion-like composition is deeply conserved across a wide diversity of eukaryotes, with some clades of proteins conserving multiple prion-like domains within their individual sequences.

Methods

Data

Sets of orthologs for the following protein families were downloaded from OrthoDB database version 10.1 at the Eukaryota level (Kriventseva et al., 2019): Pub1/Tia1 (OrthoDB group 1066369at2579), Nam8/Ngr1 (group 775799at2579) and Pab1 (group 1027234at2759). These data also contain paralogs. In addition, lists of reciprocal-best-hits (RBH) orthologs were made for the same proteins across sets of reference eukaryotic proteomes downloaded from UniProt version 2019_1 (UniProt Cosortium, 2019). These were cross-referenced with the OrthoDB sets to determine the subset of OrthoDB annotations that are such RBH orthologs. Also, only OrthoDB proteins that are from the UniProt reference proteomes were considered for further analysis. FASTA-format files of the Pub1/Tia1/Tiar and Nam8/Ngr1 families of proteins are provided as Supplemental Data (Datas S1 and S2).

Multiple sequence alignment and phylogenetic trees

Multiple sequence alignments were constructed using KMAD (Lange, Wyrwicz & Vriend, 2016), using default parameters. KMAD is designed to handle sequences with intrinsic disorder, a trait expected characteristically of prion-like proteins (Harbi & Harrison, 2014b). Phylogenetic trees were made using PHYML with Bayesian information criterion and aBayes branch support (Guindon et al., 2005), with the exception of a tree including extra paralogs described below. The aBayes branch support is a Bayesian-like transformation of the standard approximate likelihood ratio test implemented in PhyML, that is very fast and has high statistical power (Anisimova et al., 2011).

To construct a phylogenetic tree with extra paralogs, we used a stepwise procedure of tree construction. Examination of BLASTP (Altschul et al., 1997) output indicates that Pab1 and its orthologs were the next most similar RRM-domain-containing proteins in yeast that are widely conserved across eukaryotes (Table S3). Pab1 is a polyadenylate-binding protein that is an important interactor at various stages of messenger mRNA stability, biogenesis and translation. We used similarities to Pab1 to help us to filter for further paralogs to the Pub1/Tia1 and Nam8/Ngr1 proteins as follows. BLASTP was used to compare each ortholog from the Pub1/Tia1, Nam8/Ngr1 and Pab1 protein sets against their own proteomes, and any significant matches (with e-value ≤1E−04) against the majority (>50% length) of a proteome sequence that matched a Pub1/Tia1 or Nam8/Ngr1 sequence in preference to a Pab1 sequence were labelled as potential paralogs of Pub1/Tia1 or Nam8/Ngr1, formed more recently in specific eukaryotic clades. Then, a phylogenetic tree was constructed using the RaxML programme (using the PROTGAMMAJTTF mutation parameters and the ‘–f a’ flag for rapid bootstrap analysis and 100 bootstrap replicates) (Stamatakis, 2015) with the Pub1/Tia1, Nam8/Ngr1 and Pab1 ortholog sets plus the additional potential paralogs proteins as input. The tree was pruned at an appropriate node to remove all of the Pab1 sequences and associated paralogs to leave just those paralogs that were formed more recently than the last common ancestor of the Pub1/Tia1 and Nam8 proteins.

Annotation of RNA-binding (RRM) domains

Positions of RRM RNA-binding domains were taken from annotations in InterPro version 66.0 from the database Pfam (El-Gebali et al., 2019; Mitchell et al., 2019). We checked these annotations against those also for the PROSITE and SMART databases (Letunic & Bork, 2018; Sigrist et al., 2010), and they differ by <1% of extra or fewer domain annotations, which gave us confidence to proceed with the Pfam set. Since there were some OrthoDB sequences that do not have Pfam annotations, and other annotations may be incomplete, we generated additional RRM domain annotations. Firstly, the sequences for the RRM domain annotations were extracted and then compared against the ASTRAL SCOPe database version 2.07 (Fox, Brenner & Chandonia, 2014) using BLASTP (Altschul et al., 1997) with e-value threshold 1E−04, to pull out the identities of protein domain structures that are for RRM domains. Then the ASTRAL SCOPe protein domains plus the sequence fragments for the RRM-domain annotations were then formed into a BLASTP database to compare against the OrthoDB sequences to augment the existing list of RRM-domain annotations (again using e-value threshold 1E−04). These were then reduced for overlap by sorting them in decreasing order of domain length and progressively flagging overlappers further down the list for deletion.

Annotation of prion-like domains and intrinsic disorder

Prion-like domains were annotated using the PLAAC programme with default parameters (Lancaster et al., 2014). Prion-like domains were counted using two thresholds for the PLAAC log likelihood ratio (LLR) score: >0.0 and ≥15.0. Any score >0.0 indicates some predicted propensity to form a prion-like domain; the second threshold of 15.0 was picked since the lowest LLR value for a known fungal prion-forming domain that was picked oput for experimental analysis is ~16.0 in the fission yeast prion-former Ctr4 (Sideri et al., 2017). The programme fLPS was used to detect compositional bias for glutamine and asparagine residues, which is a bias characteristic of the substantial majority of the known prion-forming domains in budding yeast and other organisms (Harrison, 2006; Harrison, 2017). Expected fractional frequencies = 0.05 for both glutamine and asparagine were used in running fLPS. fLPS Binomial P-values indicate the degree of compositional bias, and here we picked the most biased subsequence for either Q or N or Q + N for each sequence. The log10 of these P-values is investigated here for all the data analysis.

To examine for multiple prion-like domains, the sequences were split up into fragments with the ends delimited by RRM domains and other protein domains (plus a 10 amino-acid-residue buffer added onto both ends), which were then input into the PLAAC programme. Prion-forming N/Q-rich domains are highly intrinsically disordered (Harbi & Harrison, 2014b). Intrinsic disorder was annotated using the IUPRED2a and DISOPRED3 programmes with default parameters (Dosztanyi, 2018; Ward et al., 2004). The total percentage of intrinsic disorder was calculated for each whole protein sequence, since there is not an intrinsic disorder score for a specific tract that can be extracted from the output of the disorder annotation programmes. Intrinsic disorder assignments by IUPRED2a and DISOPRED3 are strongly correlated (R = 0.43 for Pub1, R = 0.47 for Nam8, P-values < 1E−20).

Results

Evolution within Saccharomycetes

In Saccharomycetes, the three main proteins from the protein families under study are Pub1, Nam8 and Ngr1. Preliminary analysis of phylogenetic trees indicated that Ngr1 formed by duplication from Nam8 in the last common ancestor of Saccharomycetes. Firstly, we assessed the conservation of prion-like composition across the Saccharomycetes clade for Pub1, and for its paralogs Nam8 and Ngr1. Prion-like composition is largely maintained for all three paralogs across this clade (Fig. 1). The substantial majority of Pub1 orthologs have PLAAC prion-like region scores ≥15.0 (53/65, 82%), and almost all >0.0 (61/65, 94%); less so for Nam8 and Ngr1 (73/114, 64% ≥15.0; 113/114, 99% >0.0).

Figure 1 Phylogenetic tree of Pub1 and Nam8 in Saccharomycetes.

This is made using PhyML (Guindon et al., 2005), as described in ‘Methods’, and drawn using Evolview (Subramanian et al., 2019). The UniProt (UniProt Cosortium, 2019) names for proteins are used. The aBayes support values are colour-coded black for <0.5, green for 0.5–0.65, magenta for 0.65–0.8 and cyan for 0.8–1.0. The PLAAC (Lancaster et al., 2014) log likelihood ratio (LLR) prion-like composition scores are depicted as a histogram around the circumference of the tree. The UniProt identifier for Pub1 is P32588, for Nam8 it is Q00539 and for Ngr1 it is P32831.

Evolution across eukaryotes

Furthermore, this prion-like composition is deeply conserved across large diverse clades within the whole eukaryotic domain for the Pub1/Tia1 and Nam8/Ngr1 families (Fig. 2). Overall, 90% of sequences have PLAAC LLR scores >0.0, with 58% ≥15.0. Specifically for Pub1, 84% have scores >0.0 and 60% have scores ≥15.0. Prion-forming N/Q-rich domains are highly intrinsically disordered (Harbi & Harrison, 2014b). There is some degree of correlation of LLR score with overall percentage annotated intrinsic disorder in the sequences, particularly for Pub1 (P < 2E−16; Fig. 3B), but some clades on the phylogenetic tree are observed that have low prion-like character and relatively high intrinsic disorder (Fig. 2).

Figure 2 Phylogenetic tree of Pub1/Tia1 and Nam8/Ngr1 for the eukaryotic domain.

This is made using PhyML (Guindon et al., 2005), as described in ‘Methods’, and drawn using Evolview (Subramanian et al., 2019). The aBayes support values are colour-coded black for <0.5, green for 0.5–0.65, magenta for 0.65–0.8 and cyan for 0.8–1.0. The UniProt (UniProt Cosortium, 2019) names for proteins are used if available, but otherwise the protein identifier from the OrthoDB database is used (Kriventseva et al., 2019). Pub1/Tia1/Tiar orthologs are coloured purple, Nam8 orthologs blue and Ngr1 green. The circles of annotations are as follows in order from inside to outside: (i) labelling of Mammalian and Saccharomycetes proteins (green and brown respectively); (ii) colour-coded bars for the Pub1 Saccharomycetes clade (magenta), mammalian Tia1 (bright green), mammalian Tiar (gold), Saccharomycetes Nam8/Ngr1 (dark purple), mammalian Nam8 (dark blue); (iii) PLAAC (Lancaster et al., 2014) LLR prion-like composition score (red); (iv) percentage of residues annotated as disordered by IUPred2a (Dosztanyi, 2018) (blue); (v) percentage of residues annotated as disordered by DISOPRED3 (Ward et al., 2004) (teal); (vi) number of RRM domains in each protein (dark green); (vii) number of prion-like domains in each protein, using the threshold of >0.0 (pink); (viii) number of prion-like domains in each protein, using the threshold of ≥15.0 (gold); (ix) log10 of the fLPS binomial P-value for compositional bias for Q and N residues (Harrison, 2017) (dark grey). The data for this figure are in Tables S2 and S3.

Figure 3 Scores for prion-like composition.

(A) Distribution of prion-like PLAAC log-likelihood ratio scores (Lancaster et al., 2014) corresponding to the data in Fig. 2, with the same sequence colour-coding except Nam8 and Ngr1 are merged into one set labelled blue (Pub1/Tia1/Tiar and Nam8 + Ngr1). The intervals of prion-like scores are labelled with the value of their midpoints, that is 5 for 0–10, etc. (B) Scatter plot of intrinsic disorder versus prion-like scores with the same sequence colour-coding, with linear regression lines fitted. The P-values are two-tailed.

Prion-like character as judged by the PLAAC programme is maintained in a similar fashion for both the Pub1/Tia1 and Nam8/Ngr1 families (Fig. 3A). PLAAC LLR scores are strongly correlated with the fLPS compositional biases for Q and N residues (R = 0.70 for Nam8, and R = 0.52 for Pub1; P < 1E−20), but substantial clades can be seen on the phylogenetic tree that have low fLPS Q/N residue bias, but high PLAAC LLR scores, such as some of the fungal clades clustering near Saccharomycetes.

The larger tree with additional paralogs whose construction is described in ‘Methods’, indicates that no further major paralogs have formed during eukaryotic speciation, since these would appear as intermittent larger blocks in the tree figure (Fig. S1). Such paralogs have however formed sporadically in evolutionarily more recent clades; some of these appear to have lost the prion-like domains, or intrinsic disorder generally (Fig. S1).

Multiple prion-like domains occur in several diverse clades

Prion-forming proteins are usually characterised as having one distinct prion-forming area (Harbi & Harrison, 2014a; Harbi et al., 2012). Likewise, prion-like proteins are usually annotated by picking out a single region with the highest adjudged propensity for forming prions; the algorithms that have been derived for annotating prion-like proteins are all based on this principle (Harrison, 2017; Lancaster et al., 2014; Ross et al., 2013; Zambrano et al., 2015). However, it is possible that multiple prion-forming regions can occur within the same protein chain. We decided to check for this in the protein families under study here, since from spot-checking they often contain more than one annotated intrinsically disordered region delimited by the structured protein domains (specifically here most often the RRM RNA-binding domain). We parsed each protein sequence into fragments delimited by the positions of protein domains and annotated these fragments for prion-like composition, as described in ‘Methods’. We find that it is not uncommon to have multiple prion-like regions in one protein chain, particulary in the Nam8/Ngr1 family (11% ≥2 prion-like regions for the Nam8/Ngr1 family and 1% for the Pub1/Tia1 family, Figs. 4B and 4C). Indeed, there are specific large clades that maintain two prion-like regions (Fig. 2). Only with a lower PLAAC LLR score threshold of >0.0 does the Pub1/Tia1 family display a sizeable minority of protein sequences with multiple prion-like regions (10% with ≥2 prion-like domains; Figs. 2, 4B and 4C). A large fraction of these extra domains with prion-like composition are internal in the sequence (46% for Pub1/Tia1 and 29% for Nam8/Ngr1 for PLAAC LLR threshold >0.0).

Figure 4 Numbers of RNA-binding and prion-like domains.

(A) The number of RNA-binding domains for the proteins in Fig. 2 (sequence colour-coding same as Fig. 3). (B and C) The number of prion-like domains for the proteins in Fig. 2 (sequence colour-coding same as Fig. 3), for a PLAAC LLR score of >0.0 and ≥15.0.

The number of RNA-binding domains is largely fixed at three

The number of RNA-binding RRM domains is largely fixed at three for both the Nam8/Ngr1 and Pub1/Tia1 families of proteins (70% having three RRM domains, with just 17 sequences having >3), but proteins with just one or two RRM domains occur frequently in some specific clades (Figs. 2 and 4A).

Discussion

Evolution across eukaryotes

Analysis of the evolution across eukaryotes of prion-like composition for the Pub1/Tia1 and Nam8 families, as judged by the PLAAC programme, indicated that such composition is deeply conserved. We can therefore surmise that this composition is important for the protein function. However there are some clades that: (i) have low prion-like character and maintain relatively high annotated intrinsic disorder; (ii) have high prion-like character as judged by PLAAC, but low bias for N and Q residues, as judged by the fLPS algorithm. Sequences from such clades may be good candidates for checking for alternative compositions for prion-like domains in Pub1 that fulfil either its prion-forming ability or its role in stress granule formation. Sequences of type (ii) above would particularly be interesting candidates for further experimental study for prion-like aggregation, since the prion-like composition is made up more from non-Q + N residues that contribute to prion-forming biases in the known S. cerevisiae cases.

Multiple prion-like domains occur in several diverse clades

The occurrence of multiple prion-like domains in proteins raises the possibility of competing prion-like aggregation by the different prion-like domains within the same protein sequence. However, a large fraction of these extra domains with prion-like composition are internal in the sequence, indicating that they would have greater difficulty forming amyloid fibrils. Also, multiple prion-like domains might affect the character of intracellular condensates, that is as either solid-like or gel-like, perhaps making them more solid-like because of the increased prion-like composition relative to other types of intrinsic disorder propensity in the equivalent regions in other orthologs (Kroschwald et al., 2018).

The number of RNA-binding domains is largely fixed at three

We found very strong conservation of a three-domain RRM RNA-binding domain pattern in both the Pub1/Tia1 and Nam8 families. Proteins with just one or two RRM domains occur frequently in a conserved pattern in some smaller clades, which indicates that these are not due to annotation errors and are real evolutionary trends (Figs. 2 and 4A). There are a handful of sequences with four or more RRM domains, but it is unclear whether these may be annotation errors, due to their sporadic occurrence (Fig. 2). One notable cluster that has either as one or two RRM domains (upper left corner of Fig. 2, near clade labelled with blue band), consists mainly of vertebrate Nam8 orthologs that are assigned function in selenocysteine biosynthesis, including TRNAU1AP (transfer RNA selenocysteine 1—associated protein 1) the human RBH ortholog of S. cerevisiae Nam8, a protein that has essential functions in both early and later steps of selenocysteine biosynthesis (Small-Howard et al., 2006). Although no human disorders have been directly linked to mutations in the encoding gene of TRNAU1AP, faults in selenocysteine biosynthesis have been linked to complex disorders of metabolism and the development of muscles, skin, the nervous system and immune system (Schweizer & Fradejas-Villar, 2016). These proteins also tend to have lower annotated intrinsic disorder content and lower prion-like composition (Fig. 2).

Conclusions

Previously, we noted that Pub1/Tia1 was unique in being linked to human disease while having prion-like character aggregation in both humans and the budding yeast S. cerevisiae. Thus, we were motivated to probe further into the evolution of the Pub1/Tia1 family and its close relatives, to gain a picture of how such a protein has evolved over deep evolutionary time since the last common ancestor of eukaryotes. We discovered that the prion-like composition of the Pub1/Tia1 prion-former family, and its closely related Nam8/Ngr1 family is widely conserved across eukaryotes, with 90% of sequences having PLAAC LLR scores >0.0, with 58% ≥15.0. Also, the number of RNA-binding domains in these sequences is widely conserved, with 70% of these sequences having three RRM domains. A sizeable minority of the proteins in these ortholog families contain multiple prion-like domains within them, particularly for certain clades of the Nam8/Ngr1 family (~11% of this family having ≥2 prion-like domains with PLAAC LLR scores ≥15.0).

Overall, these results suggest that a basic scaffold comprising three RNA-binding domains and at least one prion-like region has been widely conserved since the last common ancestor of eukaryotes, and provide some support for the notion that prion-like aggregation may be an anciently conserved phenomenon for certain specific proteins.

Supplemental Information

Supplemental Information 1 Protein sequences for the Pub1/Tia1 family used for phylogenetic analysis.

Click here for additional data file.

Supplemental Information 2 Protein sequences for Nam8/Ngr1 family used for phylogenetic analysis.

Click here for additional data file.

Supplemental Information 3 Phylogenetic tree with additional paralogs.

This is made using RaxML (Stamatakis, 2015), as described in “Methods”, and drawn using iTOL (Letunic & Bork, 2019). The bootstrap values are colour-coded with gradient colouring for 0.5–1.0, ranging from red for 0.5 to green for 1.0, with <0.5 black. The UniProt (UniProt, 2019) accession numbers for proteins are used if available, but otherwise the protein identifier from the OrthoDB database is used (Kriventseva et al., 2019). Pub1/Tia1/Tiar orthologs are coloured purple, Nam8 orthologs blue and Ngr1 green. The identifiers of additional paralogs are coloured yellow. The circles of annotations are as follows in order from inside to outside: (i) PLAAC (Lancaster et al., 2014) LLR prion-like composition score (red); (ii) percentage of residues annotated as disordered by IUPred2a (Dosztanyi, 2018) (blue).

Click here for additional data file.

Supplemental Information 4 Similarities to other RNA-binding proteins in yeast proteome.

BLASTP output showing the most similar proteins to Pub1, Ngr1 and Nam8 in the yeast proteome.

Click here for additional data file.

Supplemental Information 5 Bioinformatics data for the Pub1 family used to annotate the phylogenetic trees.

Click here for additional data file.

Supplemental Information 6 Bioinformatics data for the Nam8 family used to annotate the phylogenetic trees.

Click here for additional data file.

Additional Information and Declarations

Competing Interests

Author Contributions

Data Availability

The authors declare that they have no competing interests.

Wan-Chun Su analysed the data, prepared figures and/or tables, and approved the final draft.

Paul M. Harrison analysed the data, authored or reviewed drafts of the paper, and approved the final draft.

The following information was supplied regarding data availability:

The sequences used to perform the phylogenetic analysis and data generated are available in Supplemental Files.

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
