# Peer review of "Deep conservation of prion-like composition in the eukaryotic prion-former Pub1/Tia1 family and its relatives"

_PeerJ, doi:10.7717/peerj.9023_

## Round 0.1 · original submission · Major Revisions

Dear Drs. Su and Harrison:

Thanks for submitting your manuscript to PeerJ. I have now received four independent reviews of your work, and as you will see, the reviewers raised some concerns about the research. Despite this, these reviewers are optimistic about your work and the potential impact it will have on research studying prion evolution. Thus, I encourage you to revise your manuscript, accordingly, taking into account all of the concerns raised by both reviewers.

While the concerns of the reviewers are relatively minor, this is a major revision to ensure that the original reviewers have a chance to evaluate your responses to their concerns. There are many suggestions, which I am sure will greatly improve your manuscript once addressed.

Please consider discussing Nam8/Ngr1 per Reviewer 1’s wishes. Please also include all of the relevant information in the figure legends to described your analyses and interpretations (mostly reviewer 4).

Therefore, I am recommending that you revise your manuscript, accordingly, taking into account all of the issues raised by the reviewers. I do believe that your manuscript will be ready for publication once these issues are addressed.

Good luck with your revision,

-joe

Reviewer 1 ·

Basic reporting

This article is clearly written, well structured and is consistent in its background presentation, description of aims, methods and results.

Experimental design

This article describes a bioinformatics-based study concluding that the prion-like sequential features of the prion-like Pub1 protein, important in stress granule assembly in S. cerevisiae and its human co-ortholog Tia1, which also features prion-like features and has been shown to play a role in certain human neurodegenerative diseases, are deeply conserved across eukaryotes, as is also the prion-like composition of their close relative Nam8/Ngr1. The methods used are the typical ones in this field, and led to solid results. The investigation is basically descriptive: it only offers the raw evidence of evolutionary persistence, without providing additional insights that might be useful to understand neurodegenerative diseases in humans, or basic mechanisms of granule formation.

Validity of the findings

no comment

Additional comments

Very little is said in the discussion about Nam8/Ngr1, which is a very interesting protein with a prion-like domain and has been found to be involved in or correlate with a number of human diseases.

·

Basic reporting

No comment.

Experimental design

No comment.

Validity of the findings

No comment.

Reviewer 3 ·

Basic reporting

Basic reporting

• Clear, unambiguous, professional English language used throughout.

The manuscript is clearly written in English.

• Intro & background to show context. Literature well referenced & relevant.

Introduction and background sound in a good shape. Literatures are well referenced.

• Structure conforms to PeerJ standards, discipline norm, or improved for clarity.

Yes.

• Figures are relevant, high quality, well labelled & described.

Figure 1, 2, and sup figure 3: it is necessary to provide tables (e.g. excels files) showing exact scores/numbers from the PLAAC program, the IUPRED2a program and the Pfam database for each protein. Someone may want to know the exact numbers of a particular protein.

Figure 4, figure legend: “The intervals of …. 5 for 1-10, etc”. This sentence is just a copy from the figure legend of Figure 3, and need to be fixed.

• Raw data supplied (see PeerJ policy).

Yes.

Experimental design

EXPERIMENTAL DESIGN

• Original primary research within Scope of the journal.

The manuscript fits to the scope of the journal.

• Research question well defined, relevant & meaningful. It is stated how the research fills an identified knowledge gap.

Not really. The authors should more explain the rationales behind these sequence analyses.

• Rigorous investigation performed to a high technical & ethical standard.

I don’t think so. The authors relied only on one program to detect the prion-like domains or IDRs. There are some other programs for these tasks. For annotation of RRM, they used only Pfam. Prosite can do the same task. Using the multiple programs/databases would increase reliability of this investigation.

• Methods described with sufficient detail & information to replicate.

Yes.

Validity of the findings

VALIDITY OF THE FINDINGS

• Impact and novelty not assessed. Negative/inconclusive results accepted. Meaningful replication encouraged where rationale & benefit to literature is clearly stated.

As describe above, this manuscript will be meaningful for other researchers when the authors provide the tables of all the scores/numbers of each protein.

• All underlying data have been provided; they are robust, statistically sound, & controlled.

The same as above.

• Conclusions are well stated, linked to original research question & limited to supporting results.

Conclusions are well stated. However, since the original research question is unknown, it is not linked to the conclusions.

• Speculation is welcome, but should be identified as such.

The lase statement in “Conclusion”: The results of the sequence analyses suggest, but not provide evidence, of prion-like aggregation of these protein families.

Additional comments

None

Reviewer 4 ·

Basic reporting

In this paper, Su et al. develop a pipeline to study prion-like conservation across homologues of Tia1/Pub1 protein family. First, they analyse its evolution inside the Saccharomycetes genus, then they extend it to all eukaryotes. They also check the degree of disorder and number of RNA-binding regions for each homologue. They reach the main conclusion that this protein family tend to evolutionary conserve prion-like behaviour, certain degree of disorder and RRM domains. Despite the technical part of the work is generally correct, it is my opinion that the results provide just incremental knowledge relative to previous studies and therefore that their potential impact in the area do not justify their publication in PeerJ.
Main Comments

1. The introduction although well-written, fails to provide context to the results section manuscript. It does not mention why authors consider disorder content or why they believe RRM domains conservation is important for function. Moreover, the description of the Nam8/Ngr1 pair is poor and unconnected. The introduction would benefit from a broader description of the Nam8/Ngr1 family or its human homologue TRNAU1AP. In this line, Pab1 protein (for which authors compare sequential identity) is not introduced either.
Phrase in lines 61-63 could is poorly connected with the adjacent context.

2. Authors do not justify why they check disorder content. They choose IUPred2 as the only methodology behind overall disorder, and the % of amino acids in the whole protein predicted to be disordered as a method to quantify it. PLAAC amino acid scoring system has a first bias towards disordered regions (highly scoring Q,N,Y,G and poorly scoring L,C,F,w,I,V). Although this is not exactly IUPred2's scoring system (for instance, IUPred2 also scores positively charged-residues), their predictions are expected to have a degree of overlap. This could be improved by applying several disorder predictors or a consensus method (like MobiDB, Piovesan et al. 2017). Furthermore, the quantification of disorder percentage for the complete protein and its relevance for the protein behaviour should be better defined and region known to have folded structures excluded from the calculations and comparisons.

3. It is not clear for unspecialized readers why the authors use the KMDA methodology for the alignments.

4. Yeast prions are modular. The manuscript will benefit from a study of the relationship between the position of RRMs and that of predicted prion-like domains. In a large majority of proteins, prion-like domains are located at the extremes of the proteins, this allows the assembly of the prion-like domains into fibrillar structures, keeping the globular domains appended to them and hanging from the fibrillar structures. It is difficult to envision how this function can be done by an internal sequence when you have more than two prion-like domains. In the lack of experimental evidence for the functional role of internal predicted prion-like domains, especially in the Tia1/Pub1 family, the relevance of these predictions remains questionable.

5. Conclusions should be improved to clearly reflect strictly what is demonstrated in the results section.

Other comments:
Where the authors refer to “Uniprot names it should be more correct to state Uniprot Accession number.
-Line 81. Authors should state for which clade they downloaded proteins from OrthoDB and their number. Authors state that “only OrthoDB proteins from the Uniprot reference fungi were considered for further analysis”, but further analysis includes proteins from all Eukarya. The paragraph should be rewritten to reflect how authors acquired and filtered homologues to Pub1/Tia1.
-OrthoDB and UniprotKB and PfamDB releases should be stated.
-"ASTRASCOP" should be changed for "Astral SCOPe". The downloaded release should be stated.
- Clarify if the LLR threshold 15 has any experimental basis or is completely arbitrary?
-Authors should explicitly state how the protein families were compared in BLASTP to conclude the most similar RMM-domain-containing protein family was Pab1. Moreover, a graph showing the top ranked conserved protein families would benefit the manuscript.

Figure 1.
-Authors do not explain what "Support values" represent.
-The support values colour code for black is missing.
-The scores next to I3GXC2 represent LLR? Should be clearly stated either at the top of the histogram axis or specified at the legend.
-Image could be more easy to understand by showing in bold Uniprot Accession numbers for Nam8, Pub1 and Ngr1.

Figure 2.
Same 2 first points as fig 1.

Figure 3
Colour code in legend is difficult to understand.
Probability value represents one or two tailed p-value?
Figure panel should contain same (or similar) size plots. And panel letters should be aligned.

Figure 4
Legend in figure 4 should be rewritten:
-Legend should state "color coding same as Figure 3", not "2", unless it is clearly stated the regrouping modifications done as it is in figure 3. Moreover, the color scheme in legend is repeated twice.
-The phrase "The intervals of prion-like scores are labelled with the value of their midpoints, i.e. 5 for 0-10, etc" does not seem to correspond with the Figure or if it does is difficult to understand what does it refer to.

Experimental design

Included in the Basic reporting

Validity of the findings

Included in the Basic reporting

Additional comments

Included in the Basic reporting

---

## Round 0.2 · accepted · Accept

Dear Drs. Su and Harrison:

Thanks for revising your manuscript based on the concerns raised by the reviewers. I now believe that your manuscript is suitable for publication. Congratulations! I look forward to seeing this work in print, and I anticipate it being an important resource for groups studying prion evolution. Thanks again for choosing PeerJ to publish such important work.

Best,

-joe

Reviewer 1 ·

Basic reporting

As in first submission.

Experimental design

As in first submission.

Validity of the findings

As in first submission.

Additional comments

The Authors have addressed my suggestion of including background information about Nam8/Ngr1.

Reviewer 3 ·

Basic reporting

The author accordingly responded to my comments and other reviewers' comments. The manuscript is ready for publication.

Experimental design

NA

Validity of the findings

NA

Additional comments

NA